# Pathways Linking Housing Inequalities and Health Outcomes among Migrant and Refugee Populations in High-Income Countries: A Protocol for a Mixed-Methods Systematic Review

**DOI:** 10.3390/ijerph192416627

**Published:** 2022-12-10

**Authors:** Kritika Rana, Andrew Page, Jennifer L. Kent, Amit Arora

**Affiliations:** 1Translational Health Research Institute, Western Sydney University, Campbelltown, NSW 2560, Australia; 2School of Health Sciences, Western Sydney University, Campbelltown, NSW 2560, Australia; 3Health Equity Laboratory, Campbelltown, NSW 2560, Australia; 4The University of Sydney School of Architecture, Design and Planning, The University of Sydney, Sydney, NSW 2008, Australia; 5Discipline of Child and Adolescent Health, The Children’s Hospital at Westmead Clinical School, Faculty of Medicine and Health, The University of Sydney, Westmead, NSW 2145, Australia; 6Oral Health Services, Sydney Local Health District and Sydney Dental Hospital, NSW Health, Surry Hills, NSW 2010, Australia

**Keywords:** housing, health, migrant, refugee, high-income countries, systematic review

## Abstract

Several high-income countries are currently experiencing an unprecedented and multifaceted housing crisis. The crisis is escalating rapidly, and its negative ramifications are shared disproportionately by migrant and refugee communities. Although housing is often cited as an important social determinant of health, the relationship between housing inequalities and health outcomes in the context of migrant and refugee populations remain under-explored, particularly in high-income countries. This paper presents a protocol for a mixed-methods systematic review which will synthesize the evidence on the key housing and health inequalities faced by migrant and refugee populations in high-income countries. It will inform the identification of pathways linking housing inequalities to health outcomes. The protocol for this systematic review was developed with guidance from the Joanna Briggs Institute (JBI) methodology for mixed-methods systematic reviews using a convergent integrated approach to synthesis and integration, and the Preferred Reporting Items for Systematic review and Meta-Analysis Protocols (PRISMA-P) statement. Quantitative, qualitative and mixed-methods studies reporting the association of housing inequalities with physical and mental health outcomes among refugee and migrant populations in high-income countries will be included. Medline, Web of Science, Embase, PsycINFO, Scopus and CINAHL will be searched for peer-reviewed literature. This will be supplemented by gray literature searches using Google Scholar, MedNar and WHOLIS. Two reviewers will independently screen and select studies, assess the methodological quality and conduct data extraction. This systematic review will elucidate the different pathways linking housing inequalities and health outcomes, which may guide the development of targeted housing and public health interventions to improve the health and wellbeing of migrant and refugee populations. The review is registered with PROSPERO (CRD42022362868).

## 1. Introduction

Adequate housing is recognized as a fundamental human right [1], with “adequate” signifying more than merely the presence of four walls and a roof [2]. Adequate housing encompasses a wide range of factors, including security of tenure, availability of services, materials, facilities and infrastructure, affordability, habitability, accessibility, location and cultural adequacy [3]. There is currently an unprecedented and multifaceted housing crisis in several high-income countries [4,5,6]. Australia, for example, has been experiencing a decline in housing affordability over the past three decades, with the nation consistently rated among the most unaffordable major housing markets across the world [7]. Despite government housing assistance, over 115,000 Australians are experiencing homelessness, and over two million Australians are currently living in unaffordable housing [8]. Housing affordability is also a major issue in the United States, with median home values having increased by 112% and median rents having risen by 61% between 1960 and 2016 [4]. Home-ownership rates have declined drastically since the great recession of 2008, and affordable quality housing has become challenging to obtain due the skyrocketing house and land prices [9]. Similarly, Europe has also been experiencing a crisis of housing affordability attributed to the global financial crisis of 2008 [10]. In the past decade, average house prices and rents in Europe have increased by over 30% and 15%, respectively, and homelessness has increased by 70% [11]. Overall, the housing affordability crisis in high-income countries such as Australia, Canada, the United Kingdom and the United States can be linked with the ratio of median house sales prices to median household incomes that has risen significantly since 1990 [12].

The negative ramifications of the housing crisis are shared disproportionately by socioeconomically disadvantaged and ethnic minority groups, and this gap is particularly stark in high-income countries [4,13]. In the United States, more than half of low-income households allocate over 50% of their income to housing [14]. Migrants represent a significant proportion of this group, with disparities further exacerbated by racial discrimination and precarious legal status [15]. More than 272 million individuals globally are classified as “international migrants”, with nearly two-thirds (176 million) residing in high-income countries [16]. Migrant and refugees residing in high-income countries have also been found to have experienced homelessness [17,18]. Furthermore, long waiting lists for social and public housing, a diminishing market of affordable private housing and experiences of discrimination increase the likelihood that migrants and refugees will encounter significant barriers in accessing appropriate, affordable and secure housing [19]. These population groups are subsequently limited to socioeconomically disadvantaged areas in hopes of accessing low-priced rental housing and acquiring social and community support, and so the cycle of cumulative disadvantage commences and continues [20]. Housing stress is a significant issue in Australia, with over half of all low-income households spending over 30% of their gross income on housing costs, making it increasingly difficult to make ends meet and have a healthier future [21]. In particular, migrants and refugees with low socioeconomic status could be experiencing greater impacts of housing stress, with research among recently arrived humanitarian migrants further revealing that they are twice as likely to struggle with housing payments relative to other Australians [22].

There is a bidirectional relationship between housing and health, especially between housing affordability and health [23,24]. Adequate, coordinated and timely support for individuals’ health and wellbeing is necessary to acquire and maintain sustainable housing. Similarly, adequate, affordable and secure housing is essential to maintain good health and wellbeing [24]. Moreover, there is overwhelming evidence on the role of housing as an essential social determinant of health [25]. The causal relationships between tangible physical housing defects and poor physical and mental health outcomes are widely acknowledged [26]. There is clear evidence of negative physical health effects of toxins within the home, cold indoor temperatures, damp and mold, along with overcrowding and safety factors [25,26]. Evidence supports the negative mental health effects arising from cold indoor temperatures, damp and mold, and overcrowding or lack of personal space [25,27]. Beyond the impacts of the tangible physical aspects of housing, the literature regarding health impacts of less tangible aspects of the housing experience, such as housing affordability, tenure stability and housing satisfaction, is relatively sparse. Nevertheless, both housing unaffordability and homelessness have negative consequences for physical as well as mental health outcomes [28]. The trade-off between housing expenses and non-shelter necessities has been acknowledged as a plausible pathway that links housing unaffordability and poor health outcomes [29]. The deterioration of health status could be attributed to the tension between housing costs and basic needs, where fewer resources are available for health-promoting necessities such as a balanced diet, maintenance of social networks and access to healthcare. Evidence also supports a direct association between homelessness and health outcomes such as higher morbidity, greater usage of acute hospital services and shorter life expectancy [30,31]. The social and economic aspects of housing, or the lack of housing itself, may play a key role in generating health disparities [32]. As housing is a critical pathway through which health disparities develop, a comprehensive understanding of the complexities of housing is essential to address both housing and health disparities [4].

Although housing is often cited as an important social determinant of health, the relationship between housing and health is less explored among migrant and refugee populations [25]. The causal pathways from housing as an infrastructure to healthcare are inherently complex, with many of these pathways neither fully conceptualized, nor empirically understood. Research suggests that the housing crisis has significant implications for the health and wellbeing of marginalized groups, such as ethnic minority groups and those with lower incomes, and the specific population groups that experience housing disparities furthermore experience significant health disparities across housing-related health outcomes [4]. Housing inequality, in particular, has often been used to describe differences in housing conditions, including physical housing quality and financial aspects such as housing affordability [33,34,35]. Housing inequality is also described as a situation experienced unevenly across the population, where key minority population groups do not have equal choices in housing compared to their majority counterparts [33]. For instance, migrant population groups have been found to be more likely to reside in poor-quality housing than non-migrants, which demonstrates housing inequalities in accessing appropriate and affordable housing [36].

Migrant and refugee populations represent one of the most marginalized and vulnerable groups who experience a range of inequalities due to their immigration status, along with facing discrimination on multiple grounds [37,38]. However, the housing and healthcare needs of migrants and refugees have largely gone unnoticed, as these population groups are often underrepresented in health research [39]. For example, there have only been two prior systematic reviews focused on understanding the relationship between housing and health among migrant and refugee populations [40,41], and both revealed significant research gaps (Appendix A; Table A1). In the systematic review by Ziersch et al. [41], which included 30 studies, only refugee and/or asylum-seeker populations were included, while studies that referred to “migrants” or “immigrants” were excluded. Another systematic review by Kaur et al. [40] included refugees, asylum seekers and undocumented migrants; however, only 18 studies utilizing a qualitative or mixed-methods design were included. Moreover, the aim of the systematic review by Kaur et al. [40] was to understand the enablers and barriers of accessing health and social services for migrants in precarious housing situations. Additionally, these reviews only included studies conducted until 2017 [41] and 2020 [40]. These gaps identify the need for an updated and high-quality systematic review to determine the association between housing inequalities and health outcomes among refugee and migrant populations in high-income countries. 

In summary, the objectives of this mixed-methods systematic review are to synthesize the evidence on the key housing and health inequalities prevalent among migrant and refugee populations in high-income countries, and to identify the pathways linking housing inequalities and health outcomes.

## 2. Methods

### 2.1. Protocol and Registration

The protocol for this systematic review has been developed with guidance from the Joanna Briggs Institute (JBI) methodology for mixed-methods systematic reviews using a convergent integrated approach to synthesis and integration, and the Preferred Reporting Items for Systematic review and Meta-Analysis Protocols (PRISMA-P) statement [42,43]. The protocol has been registered with the PROSPERO International Prospective Register of Systematic Reviews (CRD42022362868). 

### 2.2. Review Question

Using the Population of interest, Intervention(s), Comparator(s), Outcome(s) and study designs (PICOS) framework [44] (Table 1), a single review question has been developed that can be addressed by both quantitative studies and qualitative studies:

Among migrant and refugee populations in high-income countries, how are housing inequalities associated with physical and mental health outcomes?

### 2.3. Inclusion Criteria

According to the United Nations High Commissioner for Refugees’ (UNHCR) definition, “migrants and refugees” will be referred to as groups of people traveling in mixed movements, who may have multiple, overlapping reasons for moving [47]. Only those migrants and refugees currently residing in high-income countries as defined by the World Bank (Gross National Income per capita of USD 13,205 or more) will be included [46].

In accordance with International Organization for Migration’s (IOM) definition, a “migrant” will be considered as any person who moves across an international border away from his or her habitual place of residence, regardless of the person’s legal status: authorized migrants for purposes such as work, family and study as well as unauthorized entrants; asylum seekers and irregular/undocumented migrants [45]. As per IOM’s definition, “refugee” will include those people who obtained refugee status or humanitarian protection, and those who are fleeing persecution or organized violence [45].

Studies utilizing quantitative designs (e.g., observational studies including cross-sectional, longitudinal, cohort; and intervention studies); qualitative designs (e.g., ethnography, grounded theory, phenomenology, and action research); and mixed-methods designs will be considered for inclusion. All empirical studies published in English language in peer-reviewed and gray literature will be considered without any restrictions on the publication date.

### 2.4. Search Strategy

The search strategy will be developed in consultation with two expert health-sciences librarians. The search terms will use a combination of specific medical subject headings (MeSH), free-text words and Boolean operators. Relevant search terms related to migrants and refugees, housing, health and the relevant high-income countries will be drafted and pretested in the Medline database. Two reviewers (K.R. and A.A.) with experience in database searching will initially conduct a pilot search on two databases. Subsequently, the two reviewers will independently complete all remaining literature searches. Table 2 outlines the preliminary Medline (OVID) search strategy developed with assistance from expert health sciences librarians. Following the finalization of the Medline (OVID) search strategy, the syntax and subject headings will be adapted to the remaining databases. A manual search of the reference lists of the eligible studies and previously published systematic reviews will also be conducted, including backward and forward citation tracking of the included studies. 

### 2.5. Information Sources

The following electronic databases will be searched, without any restriction on publication date (i.e., from the time of database inception to present): Medline (OVID), Web of Science (ISI), Embase (OVID), PsycINFO, Scopus and Cumulative Index to Nursing and Allied Health Literature (CINAHL) (EBSCO). The electronic database search will be supplemented by gray literature searches using Google Scholar, MedNar and WHOLIS.

### 2.6. Study Selection

Studies identified through the electronic databases, gray literature databases and manual searches will be uploaded and collated into EndNote 20 reference management software (Clarivate Analytics, Philadelphia, PA, USA). The references will be exported to Covidence systematic review management software (Veritas Health Innovation, Melbourne, Australia) [48] and duplicates removed. Between the two reviewers (K.R. and A.A.), a calibration exercise will be performed on a pilot group of studies to refine the screening questions and ensure consistency across reviewers for screening and selecting eligible studies. The two reviewers (K.R. and A.A.) will independently screen the titles and abstracts of the identified studies according to the inclusion criteria. Articles that meet the inclusion criteria, as well as those articles that require further assessment, will be retrieved in full texts. The two reviewers will independently assess the full-text articles against the inclusion criteria to determine their eligibility for inclusion. If required, the study authors will be contacted to seek additional information to better assess its eligibility. Reasons for excluding full-text studies will also be recorded and reported in the systematic review. The reviewers will resolve disagreements through a consensus-based decision, or if necessary, discussions with a third reviewer (J.L.K.). The study selection process will be presented in the format of a Preferred Reporting Items for Systematic Reviews and Meta-Analysis (PRISMA) flow diagram [43].

### 2.7. Assessment of Methodological Quality

The quantitative and qualitative studies (and quantitative/qualitative component of mixed-methods studies) selected for retrieval will be assessed by two independent reviewers (K.R. and A.A.) for methodological validity prior to inclusion in the review using standardized critical appraisal instruments from JBI [42]. For mixed-methods studies, the relevant JBI quantitative and qualitative tools will be used. Where required, authors of studies will be contacted to request missing or additional data needed for clarification to better assess the methodological quality. If a response is not received after two contact attempts, the study will be assessed based on its available information. Any disagreements that arise between the reviewers will be resolved through a consensus-based decision, or if necessary, discussions with a third reviewer (A.P.). The results of critical appraisal will be reported in narrative form and in a table and will include information on the methodological quality issues present in each study, as well as its influence on the interpretation of the study results. All studies, regardless of the results of their methodological quality, will undergo data extraction and synthesis, where possible.

### 2.8. Data Extraction

Standardized data extraction templates have been created for quantitative and qualitative studies (Appendix A). The data-extraction forms will be pilot tested on three studies (quantitative, qualitative and mixed-methods designs) and refined to ensure that all relevant data are captured. A calibration exercise will also be performed to ensure consistency across reviewers. Quantitative and qualitative data will be extracted from studies included in the review by two independent reviewers (K.R. and A.A.). The data extracted from quantitative studies will include information on study details, population and setting, study aims, exposure and outcomes/measures in the study, statistical methods and results/effect estimates, and the author’s conclusions and reviewer’s comments (Table A2). The data extracted from qualitative studies will include information on study details, population and setting, study design and methods, study aims, main themes and subthemes/narrative description and author’s conclusions and reviewer’s comments (Table A3). 

Any additional information considered to be relevant will be recorded, and the data-extraction form will be modified accordingly. Where required, the authors of the studies will be contacted to request missing or additional data needed for data extraction. If a response is not received after two contact attempts, data extraction for the study will be conducted based on its available information. Any disagreements that arise between the reviewers will be resolved through a consensus-based decision, or if necessary, discussions with a third reviewer (A.P.). The extracted data will be presented in narrative form and in a table.

### 2.9. Data Transformation

Following data extraction, the quantitative data will be converted into “qualitized data”. This will involve transformation into textual descriptions or narrative interpretation of the quantitative results in order to respond directly to the review question.

### 2.10. Data Synthesis and Integration

As this review will follow a convergent integrated approach to synthesis and integration according to the JBI methodology for mixed-methods systematic reviews [42], the “qualitized data” will be assembled with the qualitative data extracted from the qualitative studies. The assembled data will be categorized and pooled together based on similarity in meaning to produce a set of integrated findings, which will be presented in the form of a line of action statements.

## 3. Discussion

This protocol for a mixed-methods systematic review provides detailed information on investigating the pathways linking housing inequalities and health outcomes among migrant and refugee populations in high-income countries. While there is abundant evidence of the housing disparities faced by migrants and refugees in high-income countries [17,18,19], research exposing the link between housing disparities and the inequitable health outcomes experienced by these vulnerable populations has not been synthesized. This systematic review will fuse the quantitative and qualitative evidence on the key housing and health inequalities prevalent among migrant and refugee populations in high-income countries and identify the pathways linking housing inequalities and health outcomes. 

The major strength of this systematic review is the updated and high-quality evidence generated by adopting rigorous methods and exhaustive literature search, which will contribute substantially to the scarce research conducted to explore the relationship between housing and health disparities among migrant and refugee populations. We acknowledge that as the inclusion criteria are limited to English-language studies, potentially relevant non-English studies conducted in high-income countries such as Spain, France and Italy may be excluded. 

The different pathways linking housing inequalities and health outcomes identified through this systematic review may guide the development of targeted housing and public health interventions to improve the health and wellbeing of migrant and refugee populations. Moreover, the research findings of this multidisciplinary review may assist policy makers across high-income countries to translate evidence into policy and practice.

## 4. Conclusions

This mixed-methods systematic review aims to synthesize the evidence on the key housing and health inequalities prevalent among migrant and refugee populations in high-income countries, and to identify the pathways linking housing inequalities and health outcomes. The findings of this systematic review may guide the development of targeted housing and public health interventions to improve the health and wellbeing of migrant and refugee populations residing in high-income countries.

## Figures and Tables

**Table 1 ijerph-19-16627-t001:** Inclusion criteria for quantitative and qualitative studies.

Parameter		Criteria
**Quantitative Studies**		
**P**	Population and setting	Migrants and refugees (as defined by the International Organization for Migration [45]) of all age groups in high-income countries (as defined by the World Bank [46])
**I**	Exposure (independent variable)	Measures of housing quality * (including tangible and non-tangible aspects). Tangible factors include (but are not limited to) housing conditions or characteristics (e.g., quality of the physical structure; over-crowding, or number of people per room; access to accommodation or residential mobility; and internal characteristics such as heating and cooling). Non-tangible factors include (but are not limited to) housing affordability, tenure stability and housing satisfaction.
**C**	Comparison	None or population subgroups with differences in measures of housing quality
**O**	Outcome (dependent variable)	Physical and mental health outcomes (e.g., physical and mental health status; prevalence of specific health conditions such as anxiety, depression and post-traumatic stress disorder)
**Qualitative Studies**		
**P**	Population and setting	Migrants and refugees (as defined by the International Organization for Migration) of all age groups
**I**	Interest	Experiences and perceptions related to the association between housing inequality and health outcomes
**Co**	Context	High-income countries (as defined by the World Bank)

* Inequalities in measures of housing quality will be used to determine housing inequalities.

**Table 2 ijerph-19-16627-t002:** Preliminary search strategy for MEDLINE (OVID) (6 November 2022).

#	Query	Results
1	(migrant* or immigrant* or emigrant* or asylum seeker* or refugee* or displaced person*).ti,ab,kw.	62,236
2	“Transients and Migrants”/ or “Emigrants and Immigrants”/ or Refugees/	38,721
3	1 or 2	71,818
4	(hous* or accommodation* or living condition* or residence* or refugee camp* or housing inequalit*).ti,ab,kw.	327,821
5	Public Housing/or Housing/or Refugee Camps/or Housing Quality/or Housing Instability/	21,457
6	4 or 5	336,304
7	(health* or wellbeing or “quality of life”).ti,ab,kw.	3,597,701
8	Health/or Public Health/or Health Inequities/or Health Status/or Health Equity/or “Quality of Life”/	436,013
9	7 or 8	3,684,836
10	3 and 6 and 9	4603
11	((high* or upper) adj5 income? adj5 (countr* or econom* or group? or nation?)).mp.	16,426
12	(America* or Andorra* or Antigua* or Aruba* or Australia* or Austria* or Barbuda* or Bermuda* or Britain or British or Baham* or Bahrain* or Barbad* or Belgium or Belgian* or Brunei* or Canada or Canadian* or Cayman Island* or Channel Island* or Chile* or Croatia* or Curacao* or Cyprus or Cyprian* or Cypriot? or Czech* or Darussalam or Denmark or Danish or England or English or Estonia* or Faroe Island* or Finland or Finnish or Finn? or France or French or German* or Gibralta* or Greece or Greek* or Greenland* or Guam* or Hong Kong* or Hungary or Hungarian* or Iceland* or Ireland or Irish or “Isle of Man” or Israel* or Italy or Italian* or Japan* or South Korea* or Kuwait* or Latvia* or Liechtenstein* or Lithuania* or Luxembourg* or Macao* or Malta or Maltese or Monaco or Nauru* or Netherlands or Dutch or New Caledonia* or New Zealand* or Northern Mariana Island* or Norway or Norwegian* or Oman* or Panama* or Poland or Polish or Portug* or Puerto Ric* or Romania* or Qatar* or Saint Kitts or San Marino or Saint Martin or Sint Maarten or Saudi Arabia* or Seychelles or Singapore* or Slovak* or Slovenia* or Spain or Spanish or Sweden or Swedish or Switzerland or Swiss or Taiwan* or Trinidad* or Tobago* or (Turks and Caicos Island*) or United Arab Emirates or United Kingdom or UK or United States or USA or Uruguay* or Virgin Island* or (western adj (countr* or econom* or nation*))).mp.	5,697,243
13	(America* or Andorra* or Antigua* or Aruba* or Australia* or Austria* or Barbuda* or Bermuda* or Britain or British or Baham* or Bahrain* or Barbad* or Belgium or Belgian* or Brunei* or Canada or Canadian* or Cayman Island* or Channel Island* or Chile* or Croatia* or Curacao* or Cyprus or Cyprian* or Cypriot? or Czech* or Darussalam or Denmark or Danish or England or English or Estonia* or Faroe Island* or Finland or Finnish or Finn? or France or French or German* or Gibralta* or Greece or Greek* or Greenland* or Guam* or Hong Kong* or Hungary or Hungarian* or Iceland* or Ireland or Irish or “Isle of Man” or Israel* or Italy or Italian* or Japan* or South Korea* or Kuwait* or Latvia* or Liechtenstein* or Lithuania* or Luxembourg* or Macao* or Malta or Maltese or Monaco or Nauru* or Netherlands or Dutch or New Caledonia* or New Zealand* or Northern Mariana Island* or Norway or Norwegian* or Oman* or Panama* or Poland or Polish or Portug* or Puerto Ric* or Romania* or Qatar* or Saint Kitts or San Marino or Saint Martin or Sint Maarten or Saudi Arabia* or Seychelles or Singapore* or Slovak* or Slovenia* or Spain or Spanish or Sweden or Swedish or Switzerland or Swiss or Taiwan* or Trinidad* or Tobago* or (Turks and Caicos Island*) or United Arab Emirates or United Kingdom or UK or United States or USA or Uruguay* or Virgin Island* or (western adj (countr* or econom* or nation*))).sh.	2,760,424
14	11 or 12 or 13	5,707,418
15	10 and 14	2687

## Data Availability

Not applicable.

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
