# Peer review of "Pathways Linking Housing Inequalities and Health Outcomes among Migrant and Refugee Populations in High-Income Countries: A Protocol for a Mixed-Methods Systematic Review"

_ijerph, 2022, doi:10.3390/ijerph192416627_

Round 1
Reviewer 1 Report
The comments to the authors are enclosed in a pdf file.

Reviewer 2 Report
The biggest challenge of the paper is that how to shed light on the linkage between housing and health inequalities faced by migrant and refugee populations in high-income countries with the context that “the relationship … remain under-explored...” as mentioned by the authors. Thus, as a review, we could only get the ideas of what has been done and what still needs to be done, but we could not get the general results of what still needs to be done.
1、the relation between housing and health inequalities need to be review deeply and detailed in the perspective of endogeneity, such as omitted variable bias, reverse causality.
2、narrow the relationship under a certain discipline, such as economics.
3、the backward and forward citations of all these research.
4、strongly suggest to get some empirical research evidences.
Round 2
Reviewer 1 Report
I thank the authors for providing a revised version of their manuscript.
I feel that the authors have appropriately addressed my comments and suggestion and considered them in the new manuscript version.
I have no additional comments for the authors.
Looking forward to reading the findings of this relevant revision.
Reviewer 2 Report
Some changes have been made according to the suggestions.